# Threshold Voltage Measurement Protocol "Triple Sense" Applied to GaN HEMTs

**Tamiris Grossl Bade *** , **Hassan Hamad, Adrien Lambert, Hervé Morel** and **Dominique Planson**

Univ Lyon, INSA Lyon, Université Claude Bernard Lyon 1, Ecole Centrale de Lyon, CNRS, UMR5005, Ampère, 69621 Villeurbanne, France; adrien.lambert@insa-lyon.fr (A.L.)
* Correspondence: tamirisgbade@gmail.com

**Abstract:** The threshold voltage instability in p-GaN gate high electron mobility transistors (HEMTs) has been brought into evidence in recent years. It can lead to reliability issues in switching applications, and it can be followed by other degradation mechanisms. In this paper, a $V_{th}$ measurement protocol established for SiC MOSFETs is applied to GaN HEMTs: the triple sense protocol, which uses voltage bias to precondition the transistor gate. It has been experimentally verified that the proposed protocol increased the stability of the $V_{th}$ measurement, even for measurements following degrading voltage bias stress on both drain and gate.

**Keywords:** threshold voltage; GaN HEMT; gate-injection transistor; hybrid-drain gate-injection transistor; cascode HEMT

## 1. Introduction

Gallium nitride (GaN)-based power transistors are good candidates for the next generation of high-efficiency and high switching frequency converters [1–5]. Among the presently commercialized semiconductors, GaN has the widest energy gap, the largest critical field, and the highest saturation velocity [6].

The most common GaN transistors today are high electron mobility transistors (HEMT) based on the 2D electron gas (2DEG) channel created by an heterojunction, usually Al-GaN/GaN [1]. Different technologies have been developed to produce normally-OFF HEMTs, and among the most commonly used and studied are the cascode structure [1], the gate-injection transistor, or GIT (also refereed to as p-GaN gate HEMT) [7], and the gate-injection transistor with hybrid drain (HD-GIT).

The p-GaN gate HEMTs present a threshold voltage ($V_{th}$) instability dependent on both the voltage bias stress level [8–12] and temperature [12,13]. This variation of $V_{th}$ is often associated with trapping mechanisms near the 2DEG channel in the region under and around the gate [14,15], but was also explained with the charging and discharging of the gate p-GaN layer, which has a semi-floating potential [16].

The fluctuation of the threshold voltage can introduce serious reliability issues, particularly in systems where a higher switching frequency is desired [1,8]. Parallel to that, some of the degradation mechanisms in GaN HEMT are accompanied by a $V_{th}$ shift [9,17], and having a well established protocol to characterize it could contribute to the research in these topics [18].

The $V_{th}$ instability phenomenon is also present in SiC MOSFETs [19–21], although due to different charging mechanisms. In SiC MOSFETs, it occurs due to the charging and discharging of gate traps, depending on the traps at the interface between the gate oxide and the semiconductor; and the border traps in the gate oxide near the interface [19,22].

However, what both these mechanisms have in common is the circulation of electric charge in and out of the gate structure, distinct as these may be. Therefore, it is possible that the $V_{th}$ measurement protocols established to deal with the instabilities in SiC MOSFETs could also have stabilizing effects on the $V_{th}$ of p-GaN gate GaN HEMTs.

The goal of this study is to determine if the $V_{th}$ measurement protocols established initially for SiC MOSFETs can be applied to GaN HEMTs. Particularly, the applicability of the triple sense $V_{th}$ measurement protocol established by the JEDEC standard JEP184 [23] will be used to study the characteristics of the $V_{th}$ shift on HEMTs.

It is important to highlight the fact that, even though the protocol has been defined for and previously tested on SiC MOSFET, there is no occurrence in the literature of this protocol being applied to GaN transistors of any kind. Moreover, the triple sense protocol can be performed in two different orders, depending on the initial gate polarization, and an experimental study can determine which is the best definition of the protocol for GaN HEMTs.

This study is based on two GaN HEMT devices commercially available, from different manufacturers. They are of different technologies: the gate-injection transistor (GIT) and the gate-injection with hybrid drain (HD-GIT). Additionally, a cascode structure is introduced in the context section.

This paper is structured as follows: Section 2 establishes the context of the study, resuming the characteristics of each device under study and discussing the $V_{th}$ instability phenomenon. Section 3 describes the measurement protocol proposed to increase the stability of $V_{th}$ measurement, and Section 4 details the results of the measurement series with and without a preconditioning protocol. The differences between these two approaches and the applicability of the proposed protocol are discussed in Section 5.

## 2. Context

The characteristics of the technologies of the devices under study are resumed in Section 2.1, and the instability of the threshold voltage of GIT HEMTs is discussed in Section 2.2.

### 2.1. The Devices under Study

In this section, a preliminary analysis is performed with three distinct technologies of normally-OFF GaN HEMTs, all commercially available: a cascode device, a gate-injection transistor (GIT) with a Schottky contact for the gate, and a hybrid-drain gate-injection transistor (HD-GIT) with an ohmic contact for the gate. The highlights of these three technologies are resumed in this section.

The cascode structure shown in Figure 1a is the most direct way to achieve a normally-off behavior on an HEMT [1]. With this structure, the conduction path is turned off by a Silicon MOSFET, and once the device is OFF the voltage $V_{SG}^{HEMT} > 0$ will close the normally-ON GaN transistor, blocking a voltage equal to the GaN HEMT gate–drain breakdown voltage.

The gate-injection transistor (GIT) is a device where an additional p-GaN layer on the gate lifts up the potential at the 2DEG channel, which enables normally-off operation [7]. Moreover, with $V_{GS} > V_{th}$ the injection of holes from the p-GaN gate to the channel increases electron density, what allows a modulation of the drain current with $V_{GS}$. One of the disadvantages of the GIT transistors is the current collapse phenomenon [24] that takes place after stressing events such as bias voltage $V_{DS}$ or hard-switching [25], and causes an increase in the effective ON state drain-source resistance of the transistors.

The GIT is also called p-GaN gate HEMT in the literature. The device studied here has a Schottky gate contact. In contrast, an ohmic contact can improve the conductivity modulation of the channel, but with increased leakage current [6].

The hybrid-drain GIT (HD-GIT) chosen for this study has an ohmic contact; its conduction state is controlled with the gate current $I_G$ instead of voltage. The hybrid drain consists of a parallel connection between a p-GaN layer and a metallic contact. This layer allows the injecting holes in the 2DEG channel under the drain, neutralizing the negative charge that leads to current collapse [26]. The structure of this transistor is given in Figure 1c.

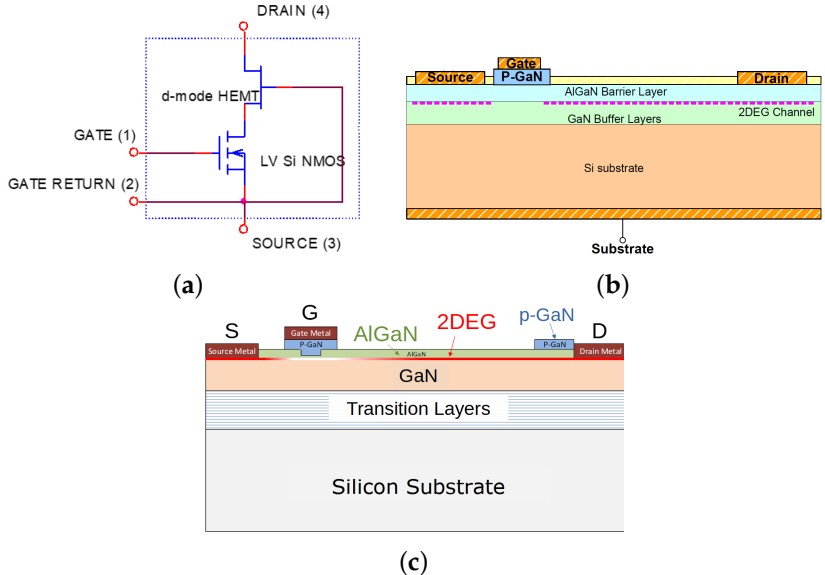

**Figure 1.** Structure of devices under study. (**a**) Cascode [27]; (**b**) GIT [28]; (**c**) HD-GIT [29].

The main characteristics of each of the three chosen devices are resumed in Table 1.

**Table 1.** Technology, manufacturer, designation, rated voltages and threshold characteristics of the devices under study (DUS). $I_D$ @ $V_{th}$: value of $I_D$ defining $V_{th}$ in the datasheet of each device.

| Technology | Designation of the DUS | $V_{th}$ | $I_D$ @ $V_{th}$ | Max $V_{GS}$ ($V$) | Max $V_{DS}$ |
|:---:|:---:|:---:|:---:|:---:|:---:|
| Cascode | ExaGaN EXA06C190LDS0 | 1.8 V | 1 mA | $[-20, 20]$ | 650 V |
| GIT | GaNsystems GS-065-030-2-L | 1.7 V | 7.5 mA | $[-7, 4]$ | 650 V |
| HD-GIT | Infineon IGT60R070D1 | 1.2 V | 2.6 mA | $[-10, (26\,\text{mA})\,^1]$ | 600 V |

[1] The HD-GIT device has an ohmic gate, which is limited by a maximum positive current instead of a positive voltage.

### 2.2. The $V_{th}$ Instability in GaN HEMTs

The $V_{th}$ instability in GIT has been brought into evidence in recent publications. Table 2 resumes the main characteristics of these instabilities as found in the present literature. The bibliography shows that both $V_{GS}$ and $V_{DS}$ static stress increase $V_{th}$. According to Tallarico et al. [16], this instability is due to the fluctuation of the electric potential of the p-GaN layer, due to the charging of this floating-point layer through hole emission and tunneling.

**Table 2.** Key points of the bibliography on dynamic $V_{th}$ in GaN HEMTs.

| Article | Device | Measurement | Type of Stress | Shifted $V_{th}$/Rated $V_{th}$ |
|:---:|:---:|:---:|:---:|:---:|
| Li 2020 [8] | GIT, in development | Keysight B1530A WGFMU (fast, (µs)) | $V_{GS}$, positive | 180%, recovery in ≈1 s |
| Oeder 2021 [9] | GIT, commercial, schottky and ohmic gates | Transfer characteristics with long measurement pulses (≈10 s) | $V_{GS}$, positive and negative | ohmic: 90%, schottky: 200% |
| Yang 2021 [10] | GIT, commercial | Fast scope (µs) | $V_{DS}$ | 150% in 4 µs, recovery in ≈2 s |

The authors could not find evidence in the literature of similar $V_{th}$ instabilities for the two other technologies studied in this paper, the HD-GIT and the cascode devices. However, Table 3 brings into evidence the fact that a standalone sweep measurement of $V_{th}$

does not give the same results as a sweep $V_{th}$ measurement following a $V_{GS}$ static stress for any of the three devices.

The data in Table 3 were obtained with a $V_{GS} = V_{DS}$ sweep measurement from 0 up to $V_{th}$, for three different preconditioning situations: with no previous stress on the gate (Sweep); following a negative bias stress; and following a positive bias stress. It is clear that the bias stress preceding the sweep measurement impacts considerably the measured $V_{th}$, due to the change in the gate static charges [9,18].

**Table 3.** $V_{th}$ measured with $V_{GS} = V_{DS}$ in three different conditions: direct sweep, with no preconditioning; following a single negative $V_{GS}$ stress; and following a single positive $V_{GS}$ stress.

|  | **Sweep** | **Negative Gate Stress** | **Positive Gate Stress** |
|---|---|---|---|
| Cascode $V_{th}$ | 1.586 V | 1.786 V | 1.787 V |
| GIT $V_{th}$ | 1.198 V | 1.600 V | 1.875 V |
| HD-GIT $V_{th}$ | 1.110 V | 1.130 V | 1.136 V |

A second problem with a single $V_{GS} = V_{DS}$ sweep measurement is the instability of the resulting $V_{th}$. Figure 2 shows the mean results of three sweep $V_{th}$ measurements following a $V_{DS} = V_{DS}^{max}$ static stress, taken with different recovery intervals. The figure shows that, for the GIT technology, the sweep measurement taken right after static stress presents a 10 % variation of $V_{th}$. This increase in the instability was not present in the HD-GIT device, but there is still a variation around 1% for each $V_{th}$ measurement

As for the cascode structure, it did not present significant variations between distinct sweep $V_{th}$ measurements following the $V_{DS}$ stress. That behavior is expected, as for the cascode the $V_{th}$ depends mainly on the Si transistor.

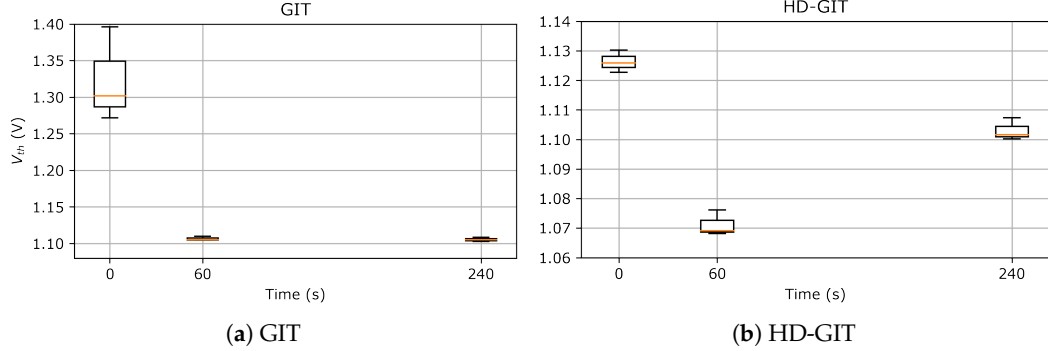

**(a)** GIT                                   **(b)** HD-GIT

**Figure 2.** Boxplot of the $V_{th}$ data measured with $V_{GS} = V_{DS}$ sweep, following a drain-source static stress ($V_{DS}$) at $t = 0$; sample size of each box: $N = 3$. Instability on $V_{th}$ results particularly for the GIT device immediately after the $V_{DS}$ stress.

This evidence indicates that $V_{th}$ measurement protocol including a preconditioning stress would be more advisable for GIT and HD-GIT technologies. To achieve this, one option would be to incorporate the $V_{th}$ measurement to a switching test circuit [10]. A second option, chosen by the authors, is to apply preconditioning stress to the gate with a device analyzer, using a measurement protocol such as those proposed by the standard JEDEC JEP184 [23] to characterize the $V_{th}$ instability in SiC MOSFETs.

From this point on, the cascode structure will not be included in the results, as its gating characteristics depend on a Si MOSFET structure instead of a p-GaN gate HEMT, and its $V_{th}$ variations are of a different nature.

Even if the $V_{th}$ shift mechanisms in GaN HEMTs are distinct from this in SiC MOSFETs, they are both linked to the charging and discharging of the gate structure through electron tunneling and trapping [16,22]. For this reason, the protocols from JEP184 [23] were studied and the "triple sense protocol" was chosen, aiming to obtain stable results on $V_{th}$

measurements as well as to build a more complete picture of the $V_{th}$ shift. The protocol is described in the next section.

As aforementioned, there is no mention in the literature of the effectiveness of the triple sense protocol on GaN HEMTs. Moreover, the standard JEDEC JEP184 [23] defines two measurement orders for the triple sense protocol, and an experimental study is necessary to determine which of these two variations is the most effective for GaN HEMTs.

## 3. Measurement Protocol

The standard JEDEC JEP184 proposes different protocols to characterize the $V_{th}$ shift in SiC MOSFETs, and among these the triple sense protocol allows the most complete description of the phenomenon. It allows the identification of four components of the $V_{th}$ instability, defined in the standard as follows [23]: the drift $V_{th}^{drift}$ is a "permanent component" of the $V_{th}$ shift, i.e., a variation from the initial measurement that remains constant on subsequent measurements; the transitory effects $V_{th}^{trans}$, and the hysteresis $V_{th}^{hyst}$ calculated as the difference between two $V_{th}$ measured after two bias stress of different polarities, and any variation on the hysteresis $\Delta V_{th}^{hyst}$. In this paper, $V_{th}^{pos}$ is threshold voltage measured following a positive gate stress, and $V_{th}^{neg}$ for a negative gate stress. Therefore, $V_{th}^{hyst} = V_{th}^{pos} - V_{th}^{neg}$.

The triple sense protocol can be established with two different combinations, represented in Figure 3a: either starting with a negative preconditioning bias stress (NBS protocol, or NBSP), or a positive preconditioning bias stress (PBS protocol, or PBSP).

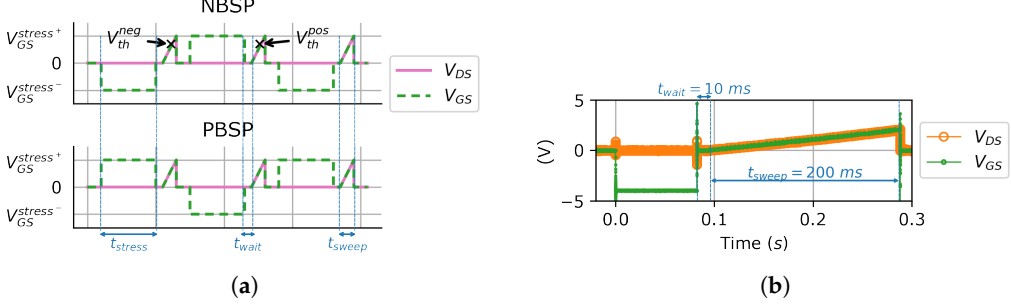

(a)　　　　　　　　　　　　　(b)

**Figure 3.** (**a**) Definition of the triple sense protocol (in accordance with [23]); the chronograph shows the preconditioning stress followed by the $V_{th}$ measurement ramp for the negative and positive bias stress protocols (NBSP and PSBP, respectively), with the definition of $V_{th}^{neg}$ and $V_{th}^{pos}$ as the measurements following a negative or positive gate stress, respectively; and (**b**) Signal measured on the the B1505A analyzer output for protocol programmed in HP BASIC; $t_{wait} = 10$ ms, $t_{sweep}$ depends on the value of $V_{th}$ measured and is in the order of hundreds of milliseconds.

The present work aims at determining which of these two combinations, NBSP or PBSP, is the most effective for GaN HEMTs.

There are four parameters on the triple sense protocol that must be adequately chosen and/or controlled: The preconditioning stress voltage level $V_{GS}^{stress}$ and time duration $t_{stress}$; the sweep measurement time duration $t_{sweep}$ and the waiting time between the preconditioning stress and the sweep measurement $t_{wait}$.

The preconditioning voltage $V_{GS}^{stress}$ can be chosen to correspond to the application gating voltages, but of course this is not mandatory. The stress should be applied for a time duration $t_{stress}$ long enough to stabilize the $V_{th}$ shift. Such a configuration allows the emulation of the "steady-sate" behavior of the gate charges [8].

However, a more thorough study would be necessary to determine values for $V_{GS}^{stress}$ and $t_{stress}$ that can be applied to GaN GITs in general, or even to optimize the characterization protocol.

The $t_{sweep}$ time should be as small as possible, because both the $V_{DS}$ and the $V_{GS}$ stress can affect the state of the charges in the gate [8,11]. The $V_{th}$ is then calculated with an interpolation of the measured $I_D(V_{GS})$.

The waiting time $t_{wait}$ should also be as small as possible, because a recovery time after the bias stress allows the gate charges to go back to their original states [8,10,12]. More importantly, $t_{wait}$ has to be the same for each measurement, to ensure that the state of the gate charges are as similar as possible during the $V_{th}$ measurement sweep.

For $t_{wait}$ to be short and adequately controlled with a Keysight B1505A, the waveforms shown in Figure 3a were programmed in HP BASIC.

The voltage level of the preconditioning stress $V_{GS}^{stress}$ was chosen by the authors to correspond to the on/off values recommended on the application notes of each device; they are summarized in Table 4. This choice was made with the goal to mimic the conditions of a device under switching stress. The stress was applied for $t_{stress} = 80$ ms for all devices, matching the order of magnitude of the protocol developed for SiC MOSFETs. As aforementioned, both $V_{GS}^{stress}$ and $t_{stress}$ can be optimized in a future study.

**Table 4.** $V_{GS}$ voltage levels chosen for the preconditioning stress $V_{GS}^{stress}$ of each transistor, corresponding to the working voltages recommended by the device manufacturers; $t_{stress} = 80$ ms for all devices.

| Preconditioning | $V_{GS}^{stress^-}$ | $V_{GS}^{stress^+}$ |
| :---: | :---: | :---: |
| GIT | $-3$ V | 6 V |
| HD-GIT | $-4$ V | 3 V |

The signal generated by the Keysight B1505A programmed in HP BASIC was measured with a scope and is plotted in Figure 3b. The $V_{th}$ ramp was programmed with a measurement step of 100 mV on $V_{GS} = V_{DS}$ in order to keep $t_{sweep}$ small, of the order of hundreds of milliseconds. This step was chosen to keep the measurement time small, to reduce the $V_{DS}$ stress over the DUS. If this stress is too long, it can compromise the stability of the characterization protocol.

Comparisons between direct and preconditioned sweep measurements are made for the same $V_{GS}$ measurement step, so that the variation measured for each of these protocols is comparable.

The measurement programmed on HP BASIC allowed a reproducible $t_{wait}$ of the order of 10 ms. The $V_{th}$ measurement is taken at the values of the drain current $I_D$ specified in the datasheet of each component, i.e., the values given in the fourth column of Table 1.

All the measurements described in the paper were performed at constant ambient temperature (25 °C). The literature indicates that lower temperatures increase the recovery of the threshold voltage shift [8]. As a result, conducting this study under ambient temperature constitutes a worst-case scenario.

## 4. Results and Discussion

The results presented in this section determine whether the triple sense protocol is also effective for GaN HEMTs, and which of the NBSP or PBSP protocols should be recommended for these devices.

The variation of the $V_{th}$ measurement results with both the PBSP and NBPS protocols are compared in Section 4.1, to establish if one guarantees more stable results then the other.

The robustness of the chosen protocol is tested after static stress on the devices; the results are described in Section 4.2.

### 4.1. Stability: NBSP vs. PBSP

The stability of the results obtained with either the NBSP and PBSP protocols is compared through three separate series of 21 triple-sense preconditioned $V_{th}$ measurements. The results for a single series are plotted for each device in Figure 4.

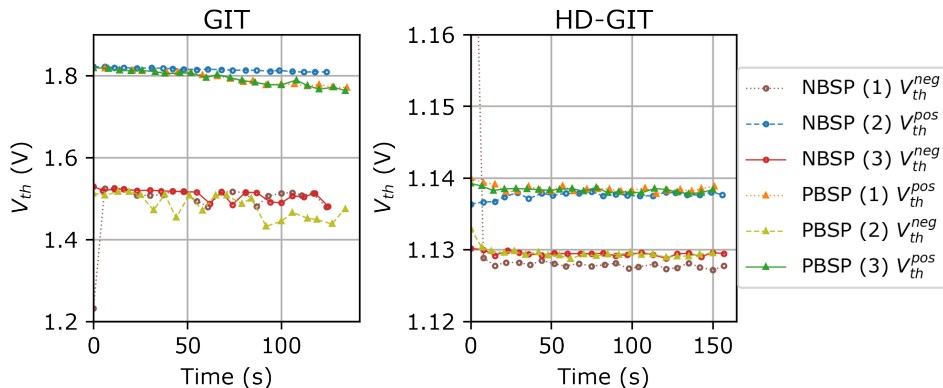

**Figure 4.** Comparison of the two triple sense protocols for the two devices under study; the data are analyzed in Table 5.

Figure 4 shows that both of the devices present a visible hysteresis $V_{th}^{hyst}$, even if for the HD-GIT device it is relatively small. The results of the GIT have higher variation of the second $V_{th}$ measurement with the PBS protocol.

To ascertain which protocol provided more stable results, the mean values and the 95 % confidence interval ($2\sigma$) of each of the $V_{th}$ measurements are resumed in Table 5. Each entry in the table is calculated from a data set of size $N = 63$ and assuming a normal distribution.

**Table 5.** Mean value and 95 % interval of confidence of each $V_{th}$ measurement. Data set size for each entry: $N = 63$.

|  |  | **GIT** | **HD-GIT** |
|---|---|---|---|
|  | $V_{th}^{neg}$ (2) | $1.555 \pm 0.032$ V | $1.129 \pm 0.001$ V |
| PBSP | $V_{th}^{pos}$ (3) | $1.863 \pm 0.008$ V | $1.138 \pm 0.001$ V |
|  | $V_{th}^{hyst}$ | $0.308 \pm 0.040$ V | $0.009 \pm 0.002$ V |
|  | $V_{th}^{pos}$ (2) | $1.869 \pm 0.005$ V | $1.138 \pm 0.001$ V |
| NBSP | $V_{th}^{neg}$ (3) | $1.565 \pm 0.015$ V | $1.130 \pm 0.001$ V |
|  | $V_{th}^{hyst}$ | $0.304 \pm 0.020$ V | $0.009 \pm 0.003$ V |

For the GIT device, the NBSP configuration proved to give more stable results than the PBSP. For the HD-GIT, there was no difference in the stability for these two protocols.

An hysteresis of 0.3 V was measured in the GIT, while the hysteresis in the HD-GIT is too small when compared to the 100 mV step of the $V_{th}$ sweep measurement, and therefore cannot be determined with sufficient accuracy. It is important to highlight that these hysteresis values could be higher if measured with fast measurements as in [8,16].

The HD-GIT device presented less $V_{th}$ hysteresis and instability because of the hybrid drain: the p-GaN layer on the drain promotes detrapping around the 2DEG channel, making the device less prone to both the current collapse and the $V_{th}$ instability.

Based on the results of this section, the authors recommend the NBS protocol as best adapted for GaN HEMTs, and will be using this version of the triple sense protocol in the rest of the paper.

### 4.2. Robustness: Reproducibility after Static Stress

To further test the reproducibility of the preconditioned $V_{th}$ measurements, experiments to ascertain the robustness of the protocol in devices under stress were performed. The NBS protocol was applied after each DUS was submitted to static voltage stress, from both $V_{GS}$ and $V_{DS}$. The results are compared with those of a simple $V_{th}$ sweep measurement for each bias stress.

The chosen values for the bias stress are summarized in Table 6. For both the GIT and the HD-GIT, the limit voltages were used. Each bias stress was applied for 1 s.

**Table 6.** Values chosen for static stress on each of the DUS; stress duration: 1 s.

|  | $V_{GS} < 0$ | $V_{GS} > 0$ | $V_{DS}$ |
|---|---|---|---|
| GIT | −4 V | 7 V | 650 V |
| HD-GIT | −10 V | 3 V | 600 V |

Three series of measurements were performed for each device and each static stress type. The results of a single series are presented in Figure 5, normalized to the first $V_{th}$ measurement ($V_{th}(t = 0)$) for each set. Therefore, Figure 5 shows the variation of the measured value of $V_{th}$ vs. time.

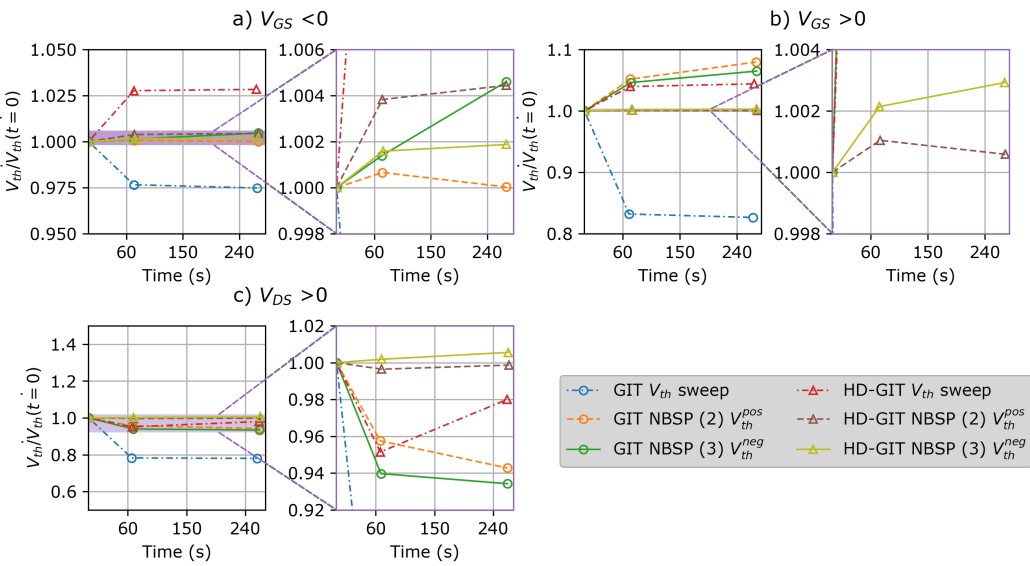

**Figure 5.** Variation of $V_{th}$ vs. time after a single static stress lasting 1 s and with amplitude as described in Table 6. The results are normalized to the first measurement, performed immediately after the first stress $V_{th}(t = 0)$. The data are further analyzed in Table 7. The figure shows that the variation on the $V_{th}^{shift}$ is higher with a direct sweep measurement.

To analyze the robustness of the $V_{th}$ protocol in more detail, Table 7 summarizes the maximal variation of $V_{th}$ for each device and type of static stress. The variations are calculated relative to $V_{th}(t = 240$ s$)$, i.e., the measurement result on the recovered device.

For the two devices, the standalone $V_{th}$ sweep measurement presented a $V_{th}$ variation up to 10 times higher than that of the preconditioned measurement. Therefore, for these two technologies, the preconditioning $V_{th}$ measurement protocol gives more robust results even after the device was submitted to bias stress.

In conclusion, the preconditioning is indispensable to obtain repeatable measurements of the $V_{th}$, because it emulates the charging state of the gate under stress in a switching application. Moreover, for the GIT technology the NBS protocol guarantees more stable measurement results even after the devices were submitted to stress.

**Table 7.** Maximal variation of $V_{th}$ relative to $V_{th}(t = 240\ s)$, analysis of the data in Figure 5. Data set size for each entry: $N = 3$. The table shows that the variation in the $V_{th}^{shift}$ is up to ten times higher with a direct sweep measurement.

|  |  | GIT | HD-GIT |
|---|---|---|---|
| $V_{GS} < 0$ | Sweep $V_{th}$ | 3.005% | 2.707% |
|  | NBSP $V_{th}^{pos}$ (2) | 0.080% | 0.482% |
|  | NBSP $V_{th}^{neg}$ (3) | 0.669% | 0.246% |
| $V_{GS} > 0$ | Sweep $V_{th}$ | 20.276% | 4.637% |
|  | NBSP $V_{th}^{pos}$ (2) | 8.530% | 0.235% |
|  | NBSP $V_{th}^{neg}$ (3) | 4.110% | 0.379% |
| $V_{DS}$ max | Sweep $V_{th}$ | 26.548% | 5.624% |
|  | NBSP $V_{th}^{pos}$ (2) | 6.749% | 0.511% |
|  | NBSP $V_{th}^{neg}$ (3) | 10.918% | 1.983% |

## 5. Discussion and Applications

The triple sense protocol first established for SiC MOSFETs presented a good performance when applied to GaN HEMTs, particularly increasing both the stability and the robustness of the measurement results for the GIT-based technologies.

Having a stable and robust $V_{th}$ measurement protocol established for GaN HEMTs will contribute to both the discussion on qualifying standards and the development of the research on the reliability and robustness of these devices. Indeed, the literature mentions degradation mechanisms in GaN HEMTs that are accompanied by a $V_{th}$ shift [9].

The triple sense protocol can help establish the medium- and long-term impact of typical stressing elements known to degrade GaN HEMTs, such as static drain-source stress in the OFF state, or hard switching [25,30,31].

An example of an application of the triple sense preconditioning protocol is given in Figure 6. The GIT device was submitted to a series of $V_{DS} = 650$ V static stress lasting 1 s, and the $V_{th}$ was measured using either the NBS protocol or the direct $V_{GS} = V_{DS}$ sweep after each stress. The $V_{th}$ values obtained with the direct sweep vary constantly, but the NBSP was able to stabilize it, again proving its robustness. It shows that the measurement of $V_{th}$ with the triple sense protocol is a more reliable method to ascertain the effect of any degrading stress on the transistor.

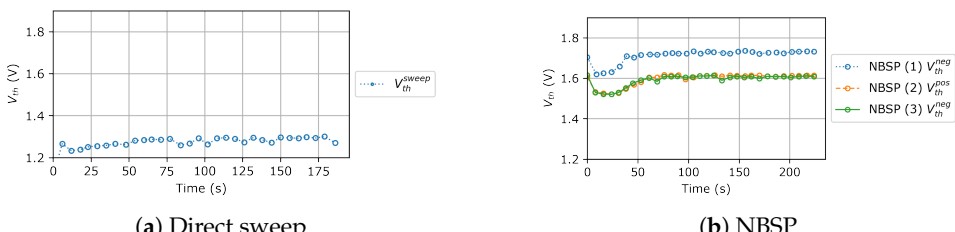

**(a)** Direct sweep        **(b)** NBSP

**Figure 6.** $V_{th}$ shift on the GIT submitted to repeated $V_{DS} = 650$ V stress; each measurement point was taken after a static stress lasting 1 s. The NBS protocol reduces fluctuation over time, particularly following a negative $V_{GS}$ stress.

The results in this study show that the $V_{GS}$ preconditioning stress included in the protocol was able to stabilize the results of the $V_{th}$ measurements. Therefore, it is proven that the triple sense protocol is very well adapted for GaN HEMTs.

These results can strongly contribute to the ongoing discussion on characterization standards for GaN HEMT devices.

As for the future work, this protocol should be put to test on devices under typical operating temperatures, to ensure these results are reproducible for temperatures higher

than 25 °C. However, the literature indicates that the increase in temperature contributes to a faster recovery of the threshold voltage shift [8]; this makes the study at ambient temperature a worst case scenario.

Additionally, the voltage level and the duration of the preconditioning stress can be optimized. Further studies are needed to determine the impact of variations on these variables. It would be interesting to compare the results presented in this paper with other proposed protocols, but unfortunately the literature on this subject is very limited at present.

## 6. Conclusions

This paper tested the applicability of the gate preconditioning protocol denominated "triple sense" from the JEDEC standard JEP184 to measure the threshold voltage of GaN HEMTs. Three commercially available devices of different normally-OFF technologies were analyzed, and the advantages of a preconditioning protocol were studies for GIT-based technologies. The triple sense protocol was proven to increase the stability of the $V_{th}$ results for GIT-based technologies, when compared to the direct sweep measurement. The results were more stable even following off-state bias stress, which indicates that the protocol has a good robustness as well. Of the two variants tested, the protocol starting and ending with a negative bias stress (NBS) improved the stability by a factor of two on the GIT device. Therefore, it is proven that the triple sense protocol can be adapted to GaN GITs even if it was initially designed for SiC MOSFETs. The protocol could be further improved by optimizing the duration and the voltage level of each preconditioning stress, and by studying the effect of temperature on the results.

**Author Contributions:** Conceptualization, H.M.; Methodology, T.G.B., H.H. and H.M.; Software, H.H.; Formal analysis, T.G.B.; Investigation, T.G.B. and A.L.; Data curation, T.G.B. and A.L.; Writing— original draft, T.G.B.; Writing—review & editing, T.G.B., H.M. and D.P.; Supervision, H.M.; Project administration, D.P.; Funding acquisition, D.P. All authors have read and agreed to the published version of the manuscript.

**Funding:** This research was funded by the Important Project of European Common Interest (IPCEI)-nano 22 of grant number 192930223.

**Data Availability Statement:** Data is contained within the article.

**Conflicts of Interest:** The authors declare no conflict of interest. The funders had no role in the design of the study; in the collection, analyses, or interpretation of data; in the writing of the manuscript; or in the decision to publish the results.

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
