# Peer review of "Threshold Voltage Measurement Protocol “Triple Sense” Applied to GaN HEMTs"

_electronics, doi:10.3390/electronics12112529_

Round 1

Reviewer 1 Report

Overall a good paper on an important topic. The discussion and findings are based on experimental results. But the figure 4 and 5 must be improved. There is a lot of space to make the figures bigger. Also use different colors for the legends. It is hard to follow with same color for NBSPs or PBSPs for different numbers. Cascode structures are basically run by the Si device. So instability of these devices are very low. This should be highlighted in section 2.1. The results from these devices should be just used as reference. Authors might remove some of the results from Cascode devices to reduce the size of the paper. The paper is too big for the contents it has. Just a suggestion. 

Author Response

The authors thank you for your review, your comments and suggestions contributed to improve the quality of our work. The modifications in the new version of the paper are highlighted in red, and the responses to your queries follow:

  • Indeed, the results on the cascode do not impact heavily on the conclusion drawn from the study. The behavior of the cascode has been further discussed in section 2.1, and the results concerning this device have been removed.

  • The figures 4 and 5 have been updated, we expect they are clearer now.

Reviewer 2 Report

This is an interesting work applying the triple-sense routine to assess threshold voltage instability in GaN HEMT. Different technologies are experimentally tested. 

Please consider that similar testing methodologies (like the double-pulse technique) are applied to normally-on HEMTs, e.g. for RF applications. I would suggest to include a discussion in the Introduction on whether the method could be applied to those devices, too.

Also, there are two aspects which, in my opinion, require a more in-depth discussion in the article. 

The first one relates to the timing of the stress. There are examples in literature using the double-pulse technique where the applied stress is much shorter than the ms-range  used here (down to ns), and which nevertheless show an evident effect of trapping. See for example: A. Santarelli et al., "Multi-bias nonlinear characterization of GaN FET trapping effects through a multiple pulse time domain network analyzer," Proc. 2015 European Microwave Integrated Circuits Conference (EuMIC). How is the timing range defined here? 

The second aspect concerns the dependency on the amplitude stress, which change the impact on trapping and on threshold voltage too. Could the author comment on some ways to decide the level of stress to be applied in a standardized procedure?

Finally, the authors evaluate the stability across a range in hundreds of seconds, whereas it has been shown that GaN might have recovery times longer than that. How the full recovery period could be defined in the procedure then, depending on the technology?

Author Response

The authors thank you for your review, your comments and suggestions contributed to improve the quality of our work. The modifications in the new version of the paper are highlighted in red, and the responses to your queries follow:

  • RF devices: We are aware that characterization protocols of either RF or power devices are often similar, however we do not have much experience on RF devices or on their differences in regard to power transistors. We do not feel capable of determining if the proposed protocol could be extrapolated to this application.

  • The duration of the preconditioning stress: this point can indeed be further studied, it is possible that the preconditioning stress duration can be reduced for GaN HEMTs.
    The duration chosen in this study was of the same order of magnitude of the one applied on SiC MOSFET characterization protocols.
    As the goal of this study is merely to verify that the protocol can be applied to GaN HEMTs, the authors did not optimize either the voltage level or the duration of the stress. This could be the object of subsequent studies.

  • The amplitude of the preconditioning stress: As mentioned in the former item, the voltage level of the preconditioning stress was not optimized. The authors chose a value corresponding to the gating voltage recommended in each of the device’s application notes, aiming to establish a protocol that would mimic the stress suffered by the devices under switching stress. however, this is absolutely not mandatory, and further studies can be conducted to determine what would be the optimal voltage level for the preconditioning stress.

  • Recovery time: Indeed, the proposed protocol does not allow the full recovery time of the device to be determined. As the main goal of the authors when establishing this protocol was to mimic hard switching conditions, the protocol aims to determine the threshold voltage of a stressed device. With this goal in view, it is not in our best interest to allow the device to fully recover from the stress.

Reviewer 3 Report

This paper presents the application details of a known threshold voltage (Vth) measurement methodology to commercial GaN HEMT DUTs. The paper is well written and adequately detailed, clearly articulating the device specs and the measurement results. However, the representation of final results can be further improved for the sake of the reader, for instance, Table 7 can be transformed instead to a bar chart for ease in understanding the final results / significance of work. Other than this - minor typos (such as repeated words, page 9, table 7 caption, 'is is') can be corrected by a thorough reading.

The reviewer has no significant major concerns. Good work !

Author Response

The authors thank you for your review, your comments and suggestions contributed to improve the quality of our work. The modifications in the new version of the paper are highlighted in red, and the responses to your queries follow:

  • Table 7 as a bar chart: indeed, that is a very good suggestion. Unfortunately we did not have the time to develop a bar chart with the complete information form the table for this round of revision. We intend submit it in the next version of the paper.

  • The text was revised to correct grammatical problems

Round 2

Reviewer 2 Report

Thanks for replying to the comments.